# COMPOSITIONAL PREFERENCE MODELS FOR ALIGNING LMS

**Dongyoung Go**
Naver Corp
Yonsei University
dongyoung.go@navercorp.com

**Tomasz Korbak**
University of Sussex
tomasz.korbak@gmail.com

**Germán Kruszewski, Jos Rozen**
Naver Labs Europe
{german.kruszewski,jos.rozen}@naverlabs.com

**Marc Dymetman**
Independent Researcher
marc.dymetman@gmail.com

## ABSTRACT

As language models (LMs) become more capable, it is increasingly important to align them with human preferences. However, the dominant paradigm for training Preference Models (PMs) for that purpose suffers from fundamental limitations, such as lack of transparency and scalability, along with susceptibility to overfitting the preference dataset. We propose Compositional Preference Models (CPMs), a novel PM framework that decomposes one global preference assessment into several interpretable features, obtains scalar scores for these features from a prompted LM, and aggregates these scores using a logistic regression classifier. Through these simple steps, CPMs allow to control which properties of the preference data are used to train the preference model and to build it based on features that are believed to underlie the human preference judgement. Our experiments show that CPMs not only improve generalization and are more robust to overoptimization than standard PMs, but also that best-of-$n$ samples obtained using CPMs tend to be preferred over samples obtained using conventional PMs. Overall, our approach demonstrates the benefits of endowing PMs with priors about which features determine human preferences while relying on LM capabilities to extract those features in a scalable and robust way.

## 1 INTRODUCTION

As the capabilities of language models (LMs) continue to advance, there is a growing need for safe and interpretable models. The dominant approach to aligning LMs with human preferences, reinforcement learning from human feedback (RLHF; Ouyang et al., 2022; Bai et al., 2022a; OpenAI, 2023), consists in training a preference model (PM) to predict human preference judgments and then finetuning an LM to maximize the reward given by the PM. However, the current PM methodology exhibits certain limitations. First, it is susceptible to overfitting the preference dataset. The PM can misrepresent human preferences by fitting to spurious correlations in its training data Gao et al. (2023). Heavily optimizing an LM against a PM incentivises the LM to exploit those flaws. This effect is known as reward hacking or Goodhart's law (Goodhart, 1984). One way of addressing reward hacking

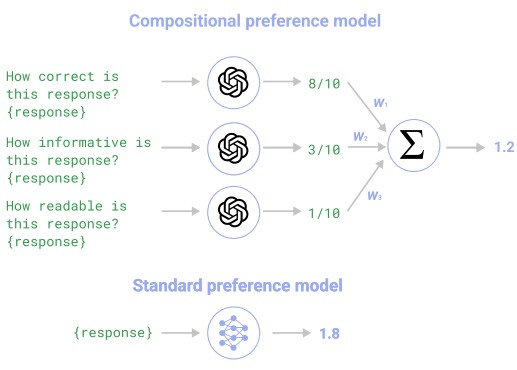

Figure 1: Compositional preference models score different features of LM responses separately and output a preference score as a linear combination of feature values.

is to impose certain inductive biases on the PM or limiting its capacity. Second, PMs are often difficult to interpret and to oversee . They project preferences onto a single scalar feature, making it difficult to know what factors are influencing their decisions. This is especially problematic for complex preferences, such as helpfulness or harmlessness, which often encompass a multidimensional combination of attributes (Bai et al., 2022a; Glaese et al., 2022; Touvron et al., 2023). Further, as LM capabilities improve, it will be increasingly harder for unassisted humans to provide feedback on LM's responses (Pandey et al., 2022; Bowman et al., 2022a). One way of addressing this problem is to use another LM to decompose those responses into simpler pieces that can be evaluated either by a human or an LM.

In this paper, we propose the Compositional Preference Model (CPM), a novel framework for learning a PM that is robust to preference model overoptimization and allows for more transparent and interpretable supervision of complex behavior. A CPM decomposes one global preference assessment into a series of simpler questions which correspond to human-interpretable features. Then, a prompted LM (e.g. GPT-3.5) is asked to assign a numerical value to each feature. Finally, the feature scores are combined into a scalar preference score using a trained logistic regression classifier.

CPMs have several advantages over standard PMs. First, they are more robust to overfitting and reward hacking. The pre-selected features on which CPMs operate provide a useful inductive bias that bootstraps learning human preferences. This, in turn, limits their vulnerability to reward hacking, as the parameter space of a PM is spanned by features selected to be meaningful and robust. Second, CPMs allow for the modular and human-interpretable supervision of complex behavior. They effectively decompose a hard question (e.g. "is this text preferable?") into a series of easier questions (e.g. "is this text easy to read?", "is this text informative?") that are easier to evaluate for an LM and easier to inspect for a human overseer. This is a simple instance of a divide-and-conquer supervision approach (Cormen et al., 2022), which recursively breaks down a problem until it is easily solvable and then combines the solutions (Irving et al., 2018; Leike et al., 2018; Christiano et al., 2018).

In our experiments, we show that CPMs generalize better and that using them results in less preference model overoptimization. Additionally, CPMs exhibit superior performance in capturing the underlying human preferences. In an auto-evaluation experiment with Claude (Anthropic, 2023) as an approximation of human evaluators (Chiang et al., 2023; Mukherjee et al., 2023; Liu et al., 2023; He et al., 2023), best-of-$n$ samples obtained using CPMs are consistently preferred over samples obtained using conventional PMs.[1]

Overall, the contributions of the paper include:

1. Introducing CPM, a novel framework for learning PMs that is more robust to overoptimization and allows for more transparent supervision, by decomposing the preference problem into a series of intuitive features linked to human preferences, and employing an LLM as a feature score extractor (Sec. 3).
2. Investigating the performance of CPMs on a diverse array of dimensions, including model robustness (Sec. 4.2), generalization (Sec. 4.3), robustness to overoptimization (Sec. 4.4), and effectiveness for preference alignment (Sec. 4.5).
3. Enabling an intuitive explanation of model optimization and generated responses (Sec. 4.6).

## 2 BACKGROUND

Let us have a dataset of comparisons $\mathcal{D} = \{x^i, y_1^i, y_2^i\}_{i=1}^N$, where $x$ is an input query and $y_1$ and $y_2$ are two possible responses to $x$, with $y_1$ the preferred response. The dominant approach to aligning language models, RLHF (Christiano et al., 2017; Ziegler et al., 2019; Ouyang et al., 2022; Bai et al., 2022a)[2], involves training a parametrized PM $R(y|x) = R_\theta(y|x)$ by defining a probability distribution

$$p_\theta(y_1 > y_2|x) \doteq \sigma(R_\theta(y_1|x) - R_\theta(y_2|x)) = (1 + \exp(R_\theta(y_2|x) - R_\theta(y_1|x))^{-1} \qquad (1)$$

and estimating $\theta$ by maximizing the likelihood of $p_\theta$ over $\mathcal{D}$. Typically $R_\theta$ is obtained by adding a scalar head on top of a base language model and fine-tuning the resulting model. Since $p_\theta$ is invariant to addition of a constant to $R_\theta$, it is standard to shift the $R$ scores such that $E_{(x,y)\sim D}[R(y|x)] = 0$.

---

[1]Code accompanying the paper is available at https://github.com/dongyoung-go/CPM

[2]CPMs can also be used with other alignment training methods both during pretraining (Korbak et al., 2023) and finetuning (Rafailov et al., 2023; Go et al., 2023).

## 3 METHOD

The Compositional Preference Model (CPM) is a multi-step approach for decomposing preference learning into individual components. We first decompose preference judgements into a set of $C$ distinct features, each designed to evaluate a specific aspect of the response $y$ (relative to context $x$). Then we use a prompted LM to assign to a pair $(x, y)$ a scalar score for each individual feature $c = 1, \ldots, C$. Finally, we employ a logistic regression classifier to combine these features into a global scalar score that best predicts the human preference judgements. This approach enables us to construct a coherent description of the characteristics that underlie these judgements.

### 3.1 FEATURE EXTRACTION USING A LANGUAGE MODEL

For each feature $c$, we consider an individual preference model $R_c$ that maps an input query $x$ and a response $y$ to a scalar score. In order to do that, we associate each feature $c$ with a specific prompt $t_c$ and compute a score $r_c = R_c(y|x, t_c)$, where $R_c$ can be a general LLM like GPT-3.5, prompted with a combination of $t_c$, $x$, and $y$. These features are designed to decompose the broad concept of preferability into a series of more straightforward and interpretable components.[3] In general, the features should be "diverse" enough so that they can cover the broad concept of preference, yet without too much "overlap" between them to decrease efficiency and interpretability. It is noteworthy that a feature can represent not only positive categories that are aligned with preferability (e.g. informativeness), but also categories that are assumed to be negatively correlated with it (e.g. biasedness). This procedure allows us to control which properties of the preference data are used to train the PM and to build it based on components that we believe to determine the human choices.

### 3.2 COMBINING MULTIPLE FEATURES

The features assessed by the prompted LM serve as distinct modules, each of which evaluates a different aspect. To combine the features into an interpretable single model, we employ logistic regression to classify the preferred response in a pairwise comparison dataset.[4]

Based on the dataset $\mathcal{D} = \{x^i, y_1^i, y_2^i\}_{i=1}^N$, we obtain a feature matrix $\{x^i, \boldsymbol{r}(y_1^i|x^i), \boldsymbol{r}(y_2^i|x^i)\}_{i=1}^N$. Here $\boldsymbol{r}(y|x) = (R_1(y|x, t_1), \ldots, R_C(y|x, t_C))$ is a feature vector with decomposed feature scores. We standardize each feature score to have average $0$ and variance $1$ within the train data. We then compute the pairwise difference of the feature vectors for each pair of responses, $\boldsymbol{r}(y_1|x) - \boldsymbol{r}(y_2|x)$, and train a logistic regression classifier with this difference to predict $1$ if $y_1$ is preferred, and $0$ if $y_2$ is preferred. In other words, the distribution $p$ is formalized as:

$$p(y_1 > y_2|x) \doteq \sigma(\langle \boldsymbol{\lambda}, \boldsymbol{r}(y_1|x) - \boldsymbol{r}(y_2|x) \rangle) = (1 + \exp(\langle \boldsymbol{\lambda}, \boldsymbol{r}(y_2|x) - \boldsymbol{r}(y_1|x) \rangle))^{-1} \quad (2)$$

where $\boldsymbol{\lambda} = (\lambda_1, \ldots, \lambda_C)$ is the vector of fitted coefficients. The coefficient $\lambda_c$ indicates the importance of the feature $c$ for predicting human preference judgements. To obtain the preference score of a single sample we simply compute $\langle \boldsymbol{\lambda}, \boldsymbol{r}(y|x) - \boldsymbol{0} \rangle = \langle \boldsymbol{\lambda}, \boldsymbol{r}(y|x) \rangle$, where $\boldsymbol{0}$ is the standardized average of the feature vector $\boldsymbol{r}(y|x)$ over the training data as explained above.

## 4 EXPERIMENTS

In this section, we empirically evaluate CPM on several aspects, including model robustness (Sec. 4.2), generalization (Sec. 4.3), robustness to overoptimization (Sec. 4.4), and effectiveness for preference alignment (Sec. 4.5). We also provide an illustrative example of CPM interpretability in Sec. 4.6.

### 4.1 EXPERIMENTAL SETUP

**Datasets** We conduct experiments on two datasets, the HH-RLHF dataset (Bai et al., 2022a) and the SHP dataset (Ethayarajh et al., 2022). Both consist of pairs of responses based on helpfulness.

---

[3]See Sharma et al. (2023) and Hosking et al. (2023) for further evidence that human preference judgements can be accurately predicted from a linear combinations of such features.

[4]Expanding pairwise comparisons to rank data is possible, following the general approach of one-vs-one (Ouyang et al., 2022).

For each dataset, in order to establish a consistent setting and control for the data size factor, we sample 20K single-turn data points.

**Features** We use 13 features: `helpfulness`, `specificity`, `intent`, `factuality`, `easy-to-understand`, `relevance`, `readability`, `enough-detail`, `biased`, `fail-to-consider-individual-preferences`, `repetitive`, `fail-to-consider-context` and `too-long`, with pre-specified prompt templates (see App. C for the description of features and prompts). We use the same set of features for both datasets; prompt templates only differ in a preamble that describes $x$ as either a conversation with an AI assistant (HH-RLHF) or a StackExchange question (SHP). We also use the length of $y$, which we find to be helpful on the SHP dataset.

**Methods** To find out the ability of an LM as a feature extractor, we explore two LMs, GPT-3.5 (`gpt-3.5-turbo-0301`) and Flan-T5-XL (3B parameters) (Chung et al., 2022), using the same features and prompt templates. We refer to the CPM models based on these extractors as CPM-GPT-3.5 and CPM-Flan-T5, respectively. To select only the most important features, we add a regularization term in logistic regression and use hyperparameters selected with 5-fold cross-validation on the training dataset.

We then compare the conventional PM to these CPMs (trained respectively as described in Sec. 2 and Sec. 3.2). For a fair comparison, we train the standard PM based on the same Flan-T5-XL model that we use for the CPMs, but with an added linear head that outputs a scalar preference score. We compare the performances of CPM-GPT-3.5 and CPM-Flan-T5 with this standard PM. Implementation details are provided in App. A.

**Best-of-$n$ sampling (BoN)** To assess the robustness of PMs to overfitting, we use Best-of-$n$ (BoN) sampling (Gao et al., 2023), a simple yet effective method that has been shown to be competitive with more advanced techniques such as reinforcement learning (Hilton & Gao, 2022). BoN abstracts away from RLHF design choices such as the details of policy optimization and provides a stable proxy for RLHF performance (Nakano et al., 2021; Gao et al., 2023).

We generate $n$ responses using an initial LM $a(x)$ and evaluate the performance of the PMs on these responses. We consider the BoN distribution $x \sim \text{BoN}(a, \text{PM}, n)$, where $n$ candidates are sampled from $a$ and $x$ is the candidate maximizing the PM score. Following Gao et al. (2023), we compare the robustness of two related PMs, $\text{PM}_A(x)$ and $\text{PM}_B(x)$, by measuring the gap between their average scores relative to samples $x$ from $\text{BoN}(a, \text{PM}_A, n)$, where typically (by construction) we have $\text{PM}_A(x) > \text{PM}_B(x)$, with the gap increasing with $n$.[5]

We generate up to 25,600 BoN responses, with 256 responses for each of 100 prompts in a held-out test set.[6] We use Flan-T5-Large (780M parameters; Chung et al., 2022) as the initial LM to generate the responses. To ensure that the performance of different PMs can be compared on the same scale across different reward models, we normalize each PM score to have average 0 and variance 1 within the training data.

### 4.2 MODEL ROBUSTNESS

Model robustness refers to the sensitivity of a predictive model to the selection of its training data (Hastie et al., 2009). Specifically, it quantifies how much the model's predictions would change if we were to train it on different subsets of the preference dataset. A model with low robustness will show poor generalization on unseen data.

To assess model robustness, we independently train two PMs for each PM method, $\text{PM}_A$ and $\text{PM}_B$, on disjoint subsets of the training data, each of size 10K. We then conduct a BoN experiment and check whether the scores of these two PMs diverge with increasing $n$. As explained above, we pick the response with highest $\text{PM}_A$ score among $n$ samples and measure the gap between the scores of $\text{PM}_A$ and $\text{PM}_B$ on that sample.[7]

---

[5]The PM used for the BoN distribution is determined by the experimental design (e.g. proxy PM in the overoptimization experiment).

[6]Due to computational constraints, we only evaluate CPM-GPT-3.5 on BoN($n \leq 16$).

[7]We tested reversing the order for building BoN distribution, and the results remained unchanged. See Fig. 8 in the Appendix.

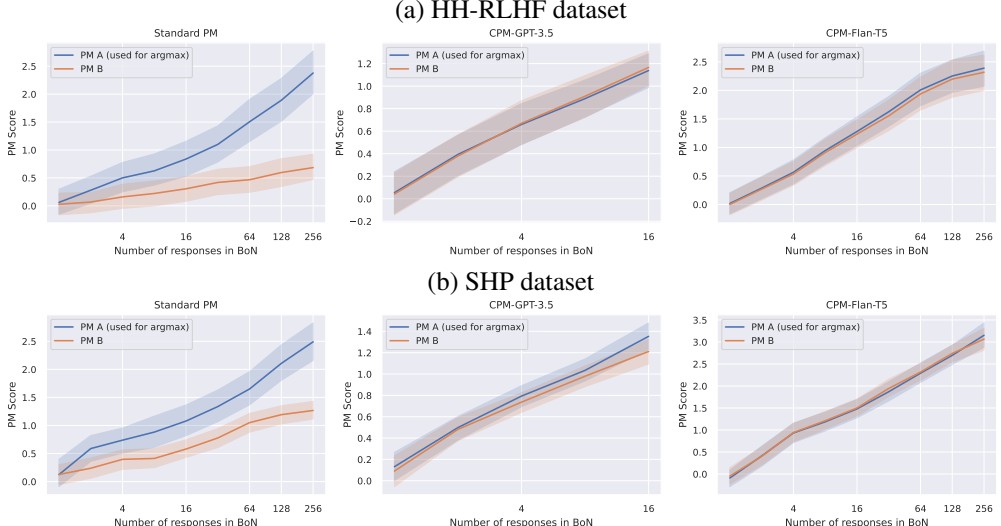

Figure 2: BoN comparison over two models fitted independently in same condition (left: Standard PM, middle: CPM-GPT-3.5, right: CPM-Flan-T5). PM A (blue line) is used for BoN selection.

Fig. 2 shows that CPM is significantly more consistent between $\text{PM}_A$ and $\text{PM}_B$ than the standard PM method in terms of the score differences, even for BoN with size 256. The smooth scaling trend as a function of $n$ suggests that our findings will generalize to larger $n$. This suggests that the small number of trainable coefficients (in this experiment 14 coefficients) makes the model robust to noise in data sampling. Still, the features extracted by LM are informative enough to build an effective preference model for alignment tuning, as we illustrate below.

## 4.3 COMPARISON WITH REFERENCE PMS

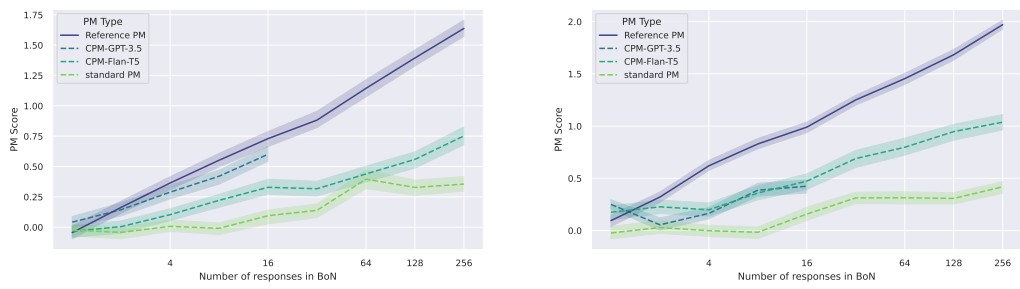

Figure 3: Comparison between PM scores relative to the distributions $\text{BoN}(a, \text{PM}_{\text{ref1}}, n)$ (HH-RLHF dataset, left) and $\text{BoN}(a, \text{PM}_{\text{ref2}}, n)$ (SHP-dataset, right).

To assess the generalizability of our CPMs, we compare them to two well-established reference PMs, $\text{PM}_{\text{ref1}}$ and $\text{PM}_{\text{ref2}}$, both instances of DeBERTa (He et al., 2020), with $\text{PM}_{\text{ref1}}$ finetuned on a large dataset including HH-RLHF[8] and $\text{PM}_{\text{ref2}}$ finetuned on a large dataset including SHP (Sileo, 2023). These PMs, trained on larger and more diverse datasets, are shown to generalize better than PMs trained on a 10K dataset (see App. B). We select BoN responses with the reference PM and then examine how their scores diverge relative to the different PMs trained on a 10K dataset as in Sec. 4.2. We hypothesize that models that diverge less from such independently trained reference PMs will generalize better to unseen data. Fig. 3 shows that all models scale monotonically with the reference PM, with the CPMs staying closer to it. This suggests that the extracted features are informative enough to allow for learning a more generalizable model of preference judgements.

---

[8]https://huggingface.co/OpenAssistant/reward-model-deberta-v3-large-v2

## 4.4 ROBUSTNESS TO OVEROPTIMIZATION

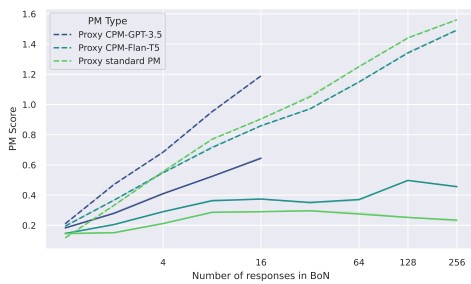 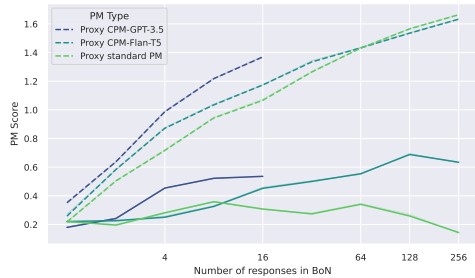

Figure 4: Overoptimization experiment in BoN distribution $\text{BoN}(a, \text{PM}_{\text{Proxy}}, n)$. Dashed line means proxy PM used for BoN selection, corresponding solid line means gold PM. (left: HH-RLHF dataset, right: SHP dataset)

Overoptimization is a type of misalignment that occurs when the preference model is overly optimized by exploiting flaws in the proxy objective (Amodei et al., 2016; Skalse et al., 2022). This can lead to the PM diverging from the true objective, which we want to optimize in alignment tuning.

To investigate overoptimization, we follow Gao et al. (2023) and construct a synthetic dataset where the output of a specific "gold" PM is assumed to be the ground truth for preferences. As gold PMs, we use reference PMs $\text{PM}_{\text{ref1}}$ and $\text{PM}_{\text{ref2}}$ (described in Sec. 4.3). We then use the gold models to generate synthetic labels to train proxy PMs using each of the studied techniques. Depending on the PM training method, overoptimizing the PM can cause it to diverge from the gold PM, which allows us to compare the robustness of different PM techniques.

Fig. 4 shows that the gap between the gold PM and the proxy PM scores increases for each PM as the candidate size $n$ increases. The distribution of the standard PM does not follow the gold PM distribution and has a larger divergence as the candidate size $n$ increases. This illustrates that fitting a standard PM can lead to overoptimization, which is consistent with existing literature (Gao et al., 2023). On the other hand, the gap between the gold and proxy PM scores is smaller for CPMs, with the gold PM score beginning to diverge later than for standard PMs. This suggests that CPMs are more robust to overoptimization. The rank correlation of the PM scores with increasing $n$ in Fig. 4, which measures this quantitatively, is provided in Table 9 in the Appendix.

## 4.5 QUALITY EVALUATION

The ultimate goal of PMs is to help align LMs with human preferences. While in the previous section we compared PMs with a certain gold PM, in this section we will investigate whether LMs aligned using CPMs are preferred by humans over LMs aligned using standard PMs. Following previous literature (Chiang et al., 2023; Mukherjee et al., 2023; Liu et al., 2023; He et al., 2023), we simulate human evaluation using a prompted LLM.

For each PM, we draw a response from $\text{BoN}(a, \text{PM}, 16)$ by generating samples from $a$ (namely Flan-T5) and selecting the best response based on the PM score. We then compare this response to vanilla Flan-T5, namely a response randomly selected from the same set of candidates. We finally use the LLM to choose which response is preferable. We refer to this metric as the "win rate". A good PM is expected to have high win rate against vanilla Flan-T5.

Importantly, we use Claude (`claude-2`; Anthropic, 2023), an LLM that was *not* used in feature extraction. Hence, we avoid *potential* subtle preference leaks from features extracted usig GPT-3.5. We use the prompt from (Chiang et al., 2023; Mukherjee et al., 2023) to rate the quality of the response selected by each PM method[9] (see Tab. 8 for the prompt used in evaluation). We perform one BoN trial with $n = 16$ for CPM-GPT-3.5 and 10 independent such trials for other PMs and report the average win rate.

---

[9]To prevent the known bias towards the first response (Chiang et al., 2023; OpenAI, 2023), we average the scores with different orderings when making a comparison.

Tab. 1 shows evaluation results. Considering that both standard PM and CPM-Flan-T5 use the same architecture and data, the higher win rate of CPM-Flan-T5 compared to standard PM suggests the advantage of decomposing preference into multiple features and using an LM as feature extractor, rather than directly using the PM based on fine-tuning the LM as in Eq. (1). CPM-GPT-3.5 shows an even higher win rate, again indicating that using a more powerful LM as feature extractor can further improve the performance of CPM.

| Win Rate | HH-RLHF | SHP |
|---|---|---|
| CPM-GPT-3.5 | **0.810** (.) | **0.672** (.) |
| CPM-Flan-T5 | 0.742 (0.034) | 0.580 (0.045) |
| Standard PM | 0.588 (0.030) | 0.564 (0.037) |

Table 1: Win rate over initial generation after BoN sampling based on each PM. Except CPM-GPT-3.5, we independently conduct 10 rounds of BoN($n = 16$) samplings and report the average win rate along with standard error.

### 4.6 MODEL INTERPRETABILITY

CPMs, as linear models, have a high degree of interpretability Hastie et al. (2009). In this section, we provide a few illustrative examples focussing on the dataset HH-RLHF.

**Coefficients** The interpretability of our model is enhanced by the fact that the feature coefficients provide a direct indication of the factors that most influence the CPM's decisions. This information can help understand the CPM's internal workings. Tab. 2 shows the top 3 largest coefficients (see Tab. 10 for full coefficients). Although the coefficients vary as they are extracted with different LMs, their orders are generally consistent, except for a few features. This observation provides some clues into how the CPM makes its decisions. In the current example, the CPM focuses on general helpfulness and also prefers responses that are detailed enough but also factually correct.

| CPM-GPT-3.5 | | CPM-Flan-T5 | |
|---|---|---|---|
| Feature | Coefficient | Feature | Coefficient |
| helpfulness | 0.246 | fail-to-consider-context | 0.420 |
| enough-detail | 0.235 | enough-detail | 0.244 |
| factuality | 0.187 | factuality | 0.227 |

Table 2: Three largest CPM coefficients on HH-RLHF dataset.

**LM-extracted features** The features extracted by the LM enable intuitive explanation of generated responses. This allows supervising complex behavior in a human-interpretable way. Tab. 3 shows examples of these features, which can be used to identify which aspects of the response contribute most to the predicted preference judgement. By decomposing a hard preference ("This text is not preferable.") into a series of easier features ("This text is generally unhelpful, as it is easy to read but has little detailed information"), it allows easier inspection for a human overseer.

## 5 ABLATION STUDY

In this ablation study, we evaluate the impact of two factors, data size and size of the LM for feature extraction on the HH-RLHF dataset.

**Data scaling trend** To investigate the effect of data size on model robustness, we hold the model size constant (3B parameters) and vary the data size used to train the PMs. We independently train each PM method on two disjoint subsets of the training data, as described in Sec. 4.2. We gradually increase the data size from 100 to 10,000. Fig. 5 shows the results of the model robustness experiment. CPMs rapidly become consistent as the data size increases and achieve stable consistency between two PMs with a data size of over 500. In contrast, standard PMs show poor consistency between models, especially when the data size is small. This suggests that CPMs are more robust than standard PMs and can produce reliable results even with a small amount of data.

**Model scaling trend** To investigate the effect of the size of the LM used for feature extraction, we gradually increase this size from Flan-T5 "small" (80M parameters) to "XL" (3B parameters) and

| Dialogue | Features | | | | |
|---|---|---|---|---|---|
| | helpfulness | readability | enough-detail | factuality | fail-to-consider-context |
| Q: What things can I create with java programming? | | | | | |
| R1: Java is a programming language that is designed to create applications that run on a computer. | 1.209 | 0.559 | 0.913 | 1.690 | 2.117 |
| R2: Robots | 0.509 | 0.559 | -0.075 | -0.569 | -0.239 |
| Q: Why is Lucifer referred to as the "morning star"? | | | | | |
| R1: Lucifer is the Morning Star because it is the brightest star in the sky. | 1.209 | 0.880 | 1.901 | 0.843 | 0.232 |
| R2: angel | -0.890 | 0.239 | -0.569 | -0.851 | -0.239 |
| Q: I'd like to evaluate some art perhaps. Can you tell me some history about Mona Lisa, the painting? | | | | | |
| R1: The Mona Lisa is a famous painting in the Louvre, Paris, France. It is attributed to Leonardo da Vinci. | 1.909 | 0.880 | 1.901 | 1.690 | 2.588 |
| R2: It was painted by Gian Lorenzo da Sica in the late 18th century. The name of the painting is inspired by the story of the famous Italian painter Leonardo da Vinci. The painting has been described as having a "poetic" quality. | 0.859 | 0.239 | 1.901 | 0.278 | -0.239 |

Table 3: Examples of feature values of CPM-GPT-3.5. Each feature value can provide information about which aspects of the response are good or bad.

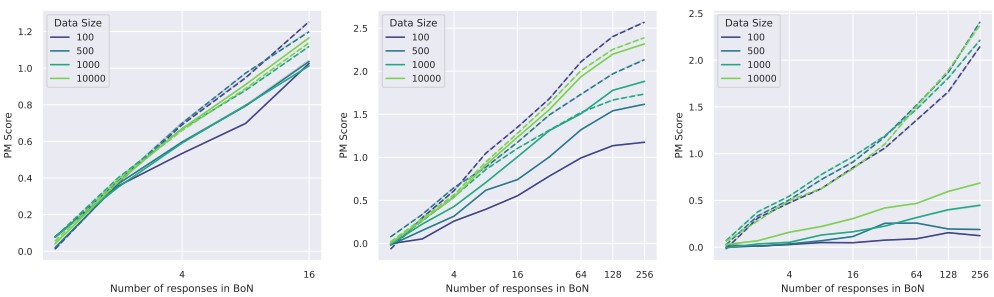

Figure 5: BoN comparison of two models fitted independently with scaling data size in HH-RLHF dataset (left: CPM-GPT-3.5, middle: CPM-Flan-T5, right: standard PM).

track two important metrics: model generalizability (described in Sec. 4.3) and win rate (described in Sec. 4.5). The training data size is fixed to 10K. As shown in Fig. 6, both model generalizability and win rate steadily improve with increasing LM size. This confirms that LM capability propagates to feature extraction, and that CPM can take advantage of it. This further means that CPMs can become even more useful as extractor LMs become more capable. The smooth and gradual increase of the win rate as a function of LM size suggests that our findings generalize to the case of using even larger LMs for feature extraction.

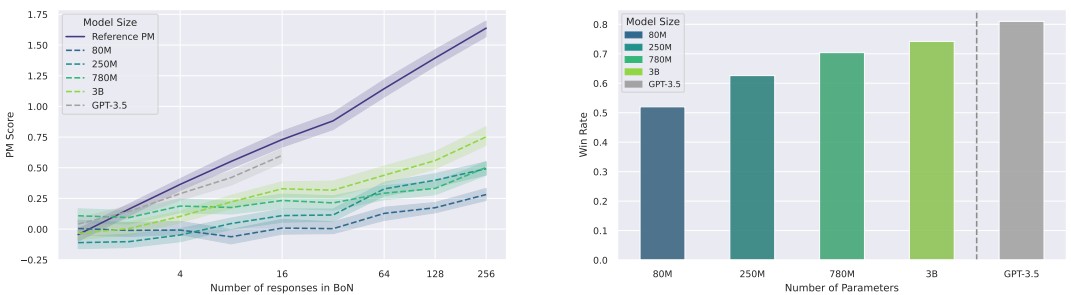

Figure 6: Model size scaling experiment using Flan-T5. (left: comparison with the reference PM, right: win rate over initial generation after BoN sampling based on each PM)

## 6 RELATED WORK

**Robustness of preference models**   PM overoptimization is an instance of reward hacking, a situation when a policy exploits flaws in its reward function (Amodei et al., 2016; Skalse et al., 2022). These flaws can come from errors of human evaluators (Pandey et al., 2022), the inherent difficulty of learning preferences of irrational agents (Mindermann & Armstrong, 2018; Shah et al., 2019) or the fragility of learned reward functions to adversarial attacks (McKinney et al., 2023). Gao et al. (2023) studied the scaling properties of PM overoptimization and Casper et al. (2023) discuss it in a broader context of open problems with RLHF. More generally, PMs can learn to be sensitive to spurious features associated with human feedback. This leads to failure modes such as sycophancy (a tendency to answer a question with a user's preferred answer, even if that answer is not correct; Cotra, 2021; Perez et al., 2022) or social bias (due narrow demographics of feedback providers; Santurkar et al., 2023; Hartmann et al., 2023). Despite its growing importance, the problem of learning robust PMs for aligning LMs is largely neglected. The present paper attempts to fill this gap.

**Decomposing tasks for LMs.**   There are numerous examples of task decomposition increasing the accuracy or robustness of language models. Breaking down problems into steps (Wei et al., 2022, chain-of-thought;) or into a sequence of subproblems depending on answers to previous subproblems (Zhou et al., 2023) are enormously beneficial for tasks involving reasoning. Others explored a stronger separation: solving subproblems independently in different LM context windows. For instance, Creswell et al. (2022) alternate between selection and inference to generate a series of interpretable, casual reasoning steps. Radhakrishnan et al. (2023) found that solving subproblems in separate context windows improves faithfulness of reasoning. Reppert et al. (2023) build compositional LM programs by applying decomposition iteratively, with a human in the loop, to facilitate science question answering. The present paper finds similar robustness benefits of decomposition for preference modeling.

**Scalable oversight**   Scalable oversight is the problem of evaluating the behaviour of agents more capable than the evaluators (Bowman et al., 2022b). On the one hand, LMs may soon grow capable of completing tasks for which humans will not be able to provide feedback. On the other, LMs might also be capable of reasoning about flaws in their evaluation procedures (Berglund et al., 2023) and exploiting them unbeknownst to overseers. Current proposals for solving scalable oversight focus on recursively relying on other LMs to assist human evaluators (Irving et al., 2018; Leike et al., 2018; Christiano et al., 2018). RL from AI feedback (Bai et al., 2022b) attempts to implement this idea by using carefully prompted LMs to generate training data for PMs. In contrast, we propose to rely on LMs during a single inference step of a PM.

## 7 CONCLUSION

We introduce Compositional Preference Models (CPMs), a simple and effective paradigm for training robust and interpretable preference models. CPMs decompose global preference scores into interpretable features and rely on language models (LMs) to extract those features. Despite their simplicity, CPMs are robust to different subsamplings of the dataset and to overoptimization, and they outperform conventional preference models at obtaining preferred best-of-$n$ samples. We believe that CPMs pave the way for combining human insights into preference judgements with the LM capabilities to extract them. Given the recent advances in LM abilities, CPMs have the potential to being used for alignment and scalable oversight of models with superhuman capabilities. One limitation of our work is that instead of a genuine human evaluation of the preferences, we use a proxy LLM (Claude 2) for the evaluation. One research direction here could be to introduce a task-oriented generation scenario (e.g. task accomplishment) where helpfulness could be evaluated easily and to understand how to inform the preference model with this scenario. Finally, another possible objective for future research would be to explore how to *elicit* decomposed features that can capture various kinds of complex preference judgements. A promising direction here would be to leverage LMs to not only score, but actually *discover* the component features that determine these judgements.

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

## A    IMPLEMENTATION DETAILS

### A.1    COMPOSITIONAL PREFERENCE MODEL

We used GPT-3.5 (`gpt-3.5-turbo-0301`) and Flan-T5-XL (3B parameters) (Chung et al., 2022) as a feature extractor, using the same features and prompt templates in Tab. 5 and Tab. 6. We excluded randomness from the generation process and selected the token with the highest likelihood.

For logistic regression classifier we used Scikit-learn (Buitinck et al., 2013). We set the choice of $L_1$ and $L_2$ regularization, weight of regularization, and solver of the logistic regression classifier as a hyperparameters and selected best hyperparameters based on 5-fold cross-validation in training dataset.

In the inference time, we made feature scores of the generated response using same LLM and templates used in training phrase. The feature scores are aggregated with the trained logistic regression classifier as described in Sec. 3.2.

### A.2    STANDARD PREFERENCE MODEL

All standard PMs were implemented using PyTorch (Paszke et al., 2019) and HuggingFace Transformers (Wolf et al., 2020) We adopt the AdamW optimizer (Loshchilov & Hutter, 2017) with $\beta = (0.9, 0.98)$ and set the weight decay to $0.01$. We conducted separate hyperparameter sweeps over learning rate and batch size for each dataset, using early-stopping based on the evaluation set with 3 steps of patience. We used a batch size of 32 and a learning rate of 1e-5 for HH-RLHF dataset and 5e-5 for SHP dataset. We used cosine learning rate schedule with 100 linear warmup steps. We used Flan-T5-XL (Chung et al., 2022, 3B parameters) for standard PMs, which is available on the Huggingface Model Hub under the model name of `google/flan-t5-xl`. Training was performed on Nvidia A100 GPU, with the longest run taking approximately 12 hours.

## B    CLAUDE EVALUATION OF THE REFERENCE PM

To evaluate the performance of reference PM in Sec.4.3 in preference alignment, we follow the same quality evaluation framework as in Sec. 4.5. Specifically, we select the best sample among 16 responses generated by Flan-T5, based on the reference PM score. We then compare this response to vanilla Flan-T5, a response randomly selected from the same set of candidates, as described in Sec. 4.5.

Again, we use Claude to rate the quality of the response selected by reference PMs (see Tab. 8 for the prompt used in evaluation). We perform 10 independent BoN trials with $n = 16$ and report the average win rate. The results are shown in Tab. 4. The reference PM outperforms CPM-Flan-T5 and standard PM. Interestingly, reference PM shows second best win rate in SHP dataset, highlighting the effectiveness in CPM-GPT-3.5. However, we note that this should be interpreted with caution as CPM-GPT-3.5 is not tested in multiple experiments.

| Win Rate | HH-RLHF | SHP |
|---|---|---|
| CPM-GPT-3.5 | 0.810      (.) | **0.672**      (.) |
| CPM-Flan-T5 | 0.742 (0.034) | 0.580 (0.045) |
| Standard PM | 0.588 (0.030) | 0.564 (0.037) |
| reference PM | **0.829** (0.050) | 0.619 (0.060) |

Table 4: Comparison of win rate between CPM and reference PM externally fitted on multiple datasets. Reference PM shows high win rate in HH-RLHF dataset, and second best win rate in SHP dataset.

## C    PROMPT USED

```
You will be given a conversation between a human and an AI assistant.
Your job is to judge whether assistant's reply {attribute_desc}.
Score that on a scale from 1 to 10 where 1 means {attr_min} and 10
means {attr_max}. Here's the conversation:

Human:
{history}

Assistant:
{reply}

Answer by outputting a number from 1 to 10 (and nothing else).

Answer:
```

Table 5: Prompt template used for HH-RLHF dataset to rate the response under given attribute.

```
You will be given a Reddit post and a reply. Your job is to judge
whether the reply {attribute_desc}. Score that on a scale from 1
to 10 where 1 means {attr_min} and 10 means {attr_max}.

POST:
{query}

Reply:
{reply}

Answer by outputting a number from 1 to 10 (and nothing else).

Answer:
```

Table 6: Prompt template used for SHP dataset to rate the response under given attribute.

## D    ADDITIONAL MODEL INTERPRETABILITY

**Feature distribution of policy**  Fig. 7 shows how the feature values of the responses change in $\text{BoN}(a, \text{CPM}, n)$ distribution with Flan-T5-Large as initial LM $a$, as a function of KL divergence $\text{KL}_{\text{BoN}}$. This simulates how the feature distribution changes as the initial policy is optimized with the current CPM and allows for inspecting which features drive reward maximization at different stages. The increase in `readability` is smaller than other features such as `enough-detail` and `fail-to-consider-context`. This means that policy shifts towards generating responses with more detailed information, rather than focusing on generating readable responses. Additionally, after a certain $n$, the slope of `fail-to-consider-context` increases, and it achieves the highest score at the end of optimization. This means that optimization pressure focuses on not failing to consider the context and avoiding unexpected lengthy responses. This decomposition makes it easier for a human to understand why the LM generated such responses.

## E    ADDITIONAL TABLES AND FIGURES

| Feature name | Attribute | Description |
|---|---|---|
| helpfulness | attribute_desc
attr_min
attr_max | is helpful for the original poster
not helpful
very helpful |
| specificity | attribute_desc
attr_min
attr_max | is specific enough
too vague
very specific |
| intent | attribute_desc
attr_min
attr_max | understands the original poster's intent
failure of understanding
perfect understanding |
| factuality | attribute_desc
attr_min
attr_max | is factually correct
egregiously incorrect
fully correct |
| easy-to-understand | attribute_desc
attr_min
attr_max | is easy to understand
very difficult to understand
very easy to understand |
| relevance | attribute_desc
attr_min
attr_max | is relevant to the original poster's question
off-topic
very relevant |
| readability | attribute_desc
attr_min
attr_max | is easy to read and not too technical for the original poster
very difficult to read
very easy to read |
| enough-detail | attribute_desc
attr_min
attr_max | provides enough detail to be helpful
too little detail
very detailed |
| biased | attribute_desc
attr_min
attr_max | is biased or one-sided
very biased
not biased at all |
| fail-to-consider-individual-preferences | attribute_desc
attr_min
attr_max | fails to consider the original poster's cultural or individual preferences
fails to consider the original poster's preferences
takes into account the original poster's preferences |
| repetetive | attribute_desc
attr_min
attr_max | is repetitive
very repetitive
not repetitive |
| fail-to-consider-context | attribute_desc
attr_min
attr_max | fails to consider the original poster's context
fails to consider the original poster's context
takes into account the original poster's context |
| too-long | attribute_desc
attr_min
attr_max | is too long
too long
not too long |

Table 7: Features and descriptions used in Tab. 5 and 6.

```
Human:
You are a helpful and precise assistant for checking the quality of
the answer. We would like to request your feedback on the performance
of two AI assistants in response to the user question displayed below.

[Question]
{query}
[The Start of Assistant 1's Answer]
{reply1}
[The Start of Assistant 2's Answer]
{reply2}

Please rate the helpfulness, relevance, accuracy, level of
details of their responses.
Each assistant receives an overall score on a scale of 1 to 10, where
a higher score indicates better overall performance.
Please first output a single line containing only two values indicating
the scores for Assistant 1 and 2, respectively. The two scores are
separated by a space. In the subsequent line, please provide a
comprehensive explanation of your evaluation, avoiding any potential
bias and ensuring that the order in which the responses were presented
does not affect your judgment.

Assistant:
```

Table 8: Prompt template to rate the writing quality of the candidate assistant model.

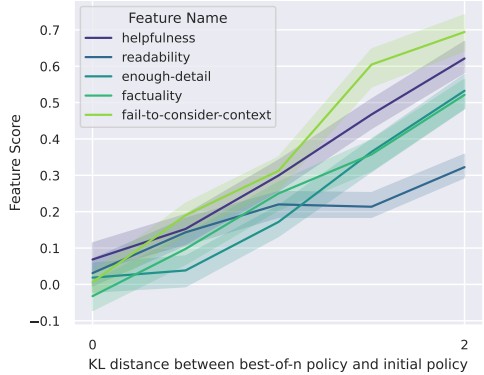 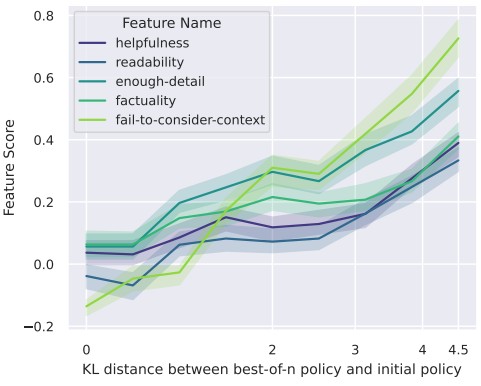

Figure 7: Feature distribution of BoN experiment (left: CPM-GPT-3.5, right: CPM-Flan-T5). Note that the $x$-axes are different. Here the KL distance of the BoN distribution from the initial distribution $a(x)$ is computed as $\mathrm{KL}_{\mathrm{BoN}} = \log n - \frac{n-1}{n}$ (Nakano et al., 2021).

|  | HH-RLHF | SHP |
|---|---|---|
| CPM-GPT-3.5 | **0.997** | **0.981** |
| CPM-Flan-T5 | 0.926 | 0.928 |
| Standard PM | 0.665 | 0.057 |

Table 9: Rank correlation between gold PM scores and proxy PM scores in BoN experiment. For each PM technique used to fit the proxy PM, we calculate and average PM scores over samples from $\mathrm{BoN}(a, \mathrm{PM}_{\mathrm{proxy}}, n)$, and compute the rank correlation between the averaged gold and proxy PM scores over different $n$.

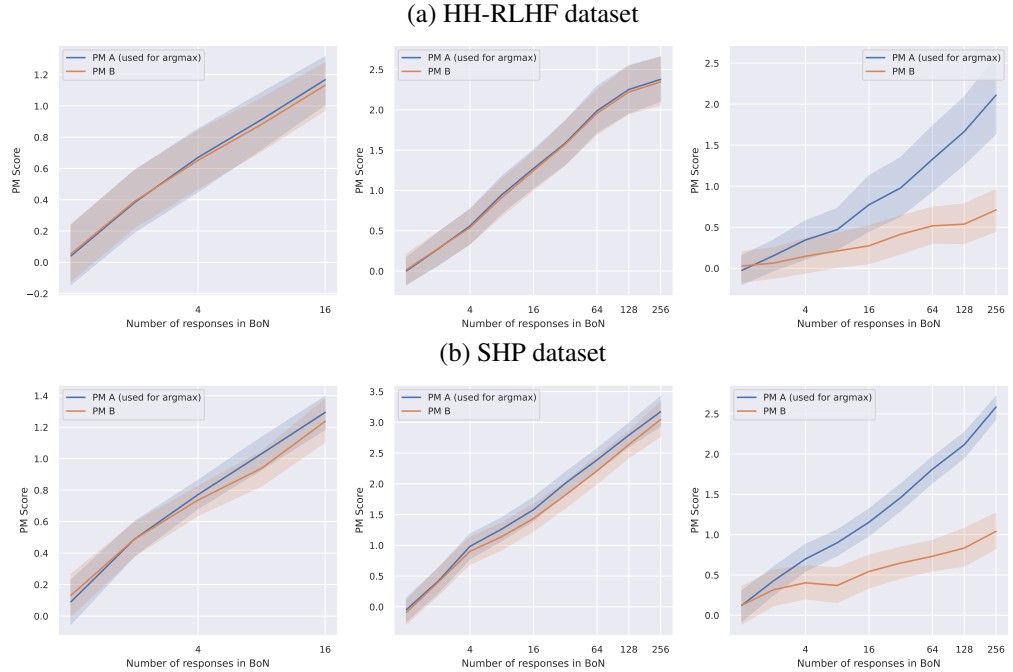

Figure 8: BoN comparison over two models fitted independently in same condition (left: CPM-GPT-3.5, middle: CPM-Flan-T5, right: standard PM) The PM A with blue line indicates the PM used for selection in BoN.

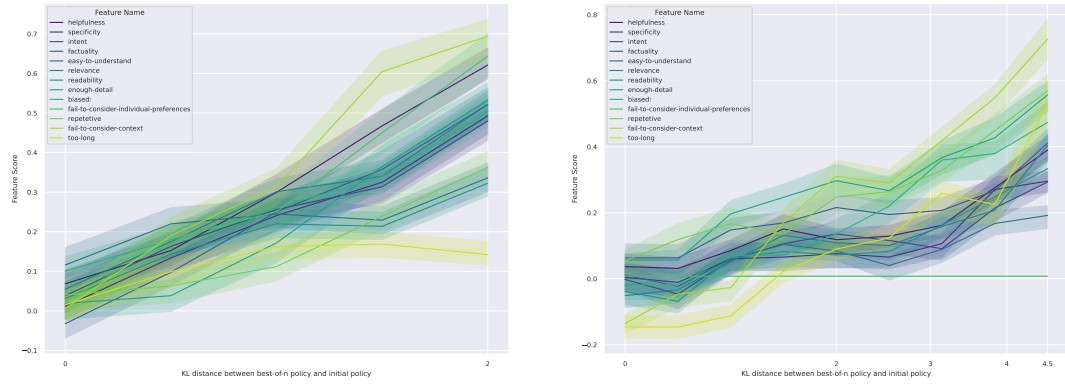

Figure 9: Feature distribution of BoN experiment (left: CPM-GPT-3.5, right: CPM-Flan-T5). Note that the $x$-axes are different. Here the KL distance of the BoN distribution from the initial distribution $a(x)$ is computed as $\text{KL}_{\text{BoN}} = \log n - \frac{n-1}{n}$ (Nakano et al., 2021).

| CPM-GPT-3.5 | | CPM-Flan-T5 | |
|---|---|---|---|
| Feature | Coefficient | Feature | Coefficient |
| helpfulness | 0.246 | fail-to-consider-context | 0.420 |
| enough-detail | 0.235 | enough-detail | 0.244 |
| factuality | 0.187 | factuality | 0.227 |
| readability | 0.184 | biased | 0.178 |
| token_length | 0.101 | easy-to-understand | 0.124 |
| specificity | 0.094 | specificity | 0.106 |
| biased | 0.086 | too-long | 0.081 |
| relevance | 0.071 | token_length | 0.075 |
| easy-to-understand | 0.069 | helpfulness | 0.037 |
| fail-to-consider-context | 0.043 | intent | 0.024 |
| too-long | 0.016 | repetitive | 0.015 |
| repetetive | 0.014 | fail-to-consider-individual-preferences | -0.042 |
| intent | -0.008 | relevance | -0.056 |
| fail-to-consider-individual-preferences | -0.056 | readability | -0.120 |

Table 10: Fitted coefficient of CPM on HH-RLHF dataset.

| | | | Features | | |
|---|---|---|---|---|---|
| Dialogue | helpfulness | readability | enough-detail | factuality | fail-to-consider-context |
| Q: why is ramon laguarta a bad ceo? | | | | | |
| R1: a bad businessman | -0.890 | -1.041 | -0.569 | -0.851 | -0.239 |
| R2: Manager | -0.890 | -1.682 | -1.063 | -0.851 | -0.710 |
| Q: What is a blizzard warning? | | | | | |
| R1: Blizzard warnings are issued by the National Weather Service to warn against snow, ice or strong winds. | 1.909 | 0.880 | 2.395 | 1.690 | 3.059 |
| R2: Blizzards can cause extremely dangerous driving conditions, with heavy snow, ice and sleet. | 1.559 | 0.88 | 1.901 | 1.408 | 2.117 |
| Q: Is climate change raising temperatures above their normal levels? | | | | | |
| R1: Climate change increases global temperature. | 1.559 | 0.559 | 1.901 | 1.690 | -0.239 |
| R2: Earth is expected to experience a warmer climate by the year 2050, the hottest on record, based on the International Solar System | 0.509 | -0.721 | 0.419 | 0.278 | -0.239 |
| Q: I'd like to evaluate some art perhaps. Can you tell me some history about Mona Lisa, the painting? | | | | | |
| R1: The Mona Lisa is a 16th century Italian portrait by Gian Lorenzo Bernini. | -0.89 | -1.041 | -0.569 | -0.851 | -0.710 |
| R2: In 1853, the artist Leonardo da Vinci created the portrait of the Madre della Signore. | -0.540 | -1.041 | -0.075 | -0.569 | -0.239 |

Table 11: Examples of feature values of CPM-GPT-3.5. Each feature value can provide information about which aspects of the response are good or bad.

## F  ADDITIONAL RESULTS RELATED TO QUESTIONS FROM REVIEWERS

### F.1  CORRELATIONS BETWEEN FEATURES

To investigate the diversity of features, we compute correlations between features in HH-RLHF dataset. The Figure 10 shows that the features have mostly positive correlations, as some of them addressing similar dimensions.

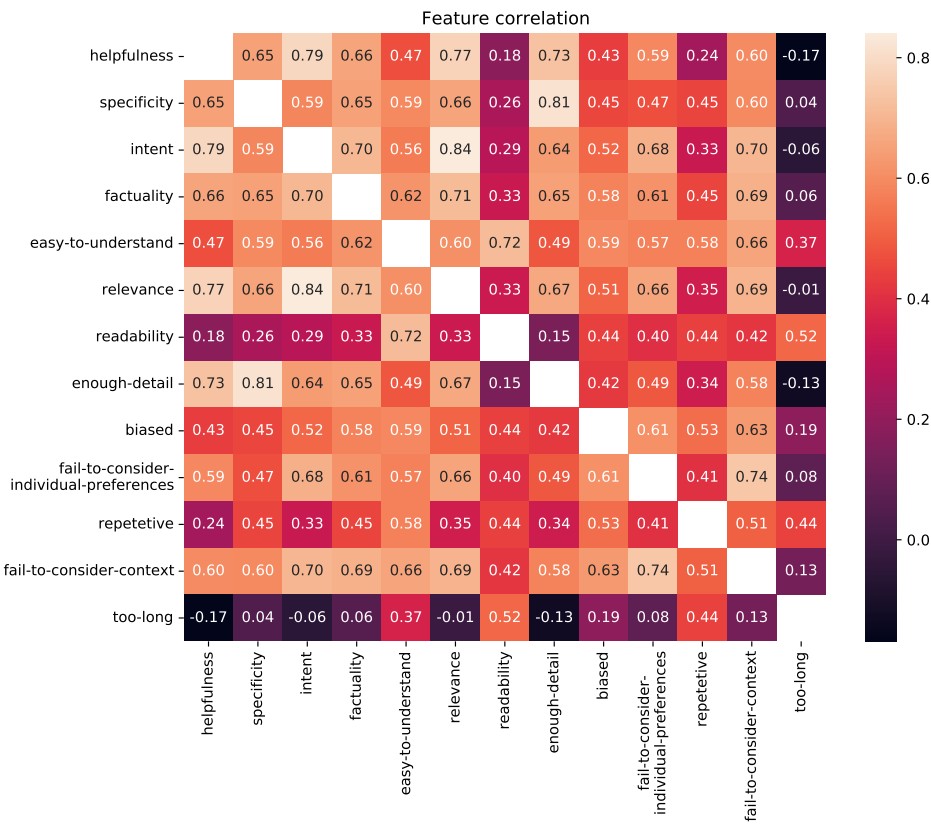

Figure 10: Full matrix of feature correlations.

### F.2  FEATURE SCALING TREND

To investigate the effect of the number $k$ of features, we gradually increase $k$ and check the win-rate of CPM-Flan-T5 with $k$ features. For this, we order the features based on their importance in Table 10, and then assess how the performance of the CPM — measured in terms of 'win-rate' quality as in Section 4.5 — varies with $k$ when we keep only the first $k$ most important features. Note that regardless of its coefficient rank, we put 'helpfulness' first in the ordered list, so that we can compare the case of "prompted PM with one holistic feature" and "compositional PM with $k$ features".

The ordered feature list is: `helpfulness, fail-to-consider-context, enough-detail, factuality, length, biased, easy-to-understand, specificity, too-long, intent, repetitive, fail-to-consider-individual-preferences, relevance, readability`. The win-rate averaged for 5 trials is described in Table 12.

The table suggests that the single holistic feature 'helpfulness' obtains a reasonable win-rate (0.707) on its own,[10] but falls short of using the combination of all features (0.742). This suggests that

---

[10]One reviewer made the interesting observation that win-rate of the prompted PM with one holistic feature 'helpfulness' still comes out ahead that of standard PM (Table 6). We hypothesize that the superior performance here of the holistic PM over the standard PM is due to the fact that our preference dataset may not be large

decomposing the features can have additional benefit for capturing the preference. Second, Table 12 shows that the performance of CPM with $k = 14$ is worse than that of CPM with $k = 6$ (0.754). This might be related to the overlap between features. However, the performance gap between $k = 14$ and $k = 6$ is small, as we employ a regularization term when fitting the logistic classifier.

| Number of features $k$ | Win Rate |
|:---:|:---:|
| $k = 1$ | 0.707 (0.030) |
| $k = 3$ | 0.715 (0.024) |
| $k = 6$ | 0.754 (0.038) |
| $k = 10$ | 0.735 (0.037) |
| $k = 14$ | 0.742 (0.034) |

Table 12: Win rate of CPM-Flan-T5 over initial generation after BoN sampling based on each PM with different number of features. We independently conduct 10 rounds of BoN($n = 16$) samplings and report the average win rate along with standard error.

### F.3 EVALUATION WITH PARAPHRASED PROMPTS

To further investigate the impact of various prompts and the robustness of the CPM's performance on prompts, we employed GPT-3.5 to paraphrase each of the original descriptions in Table 7, resulting in Table 13.

We evaluated the CPM's performance based on this second table, using the 'win-rate' quality metric described in Section 4.5. The average win rate of CPM-Flan-T5 across five independent trials was 0.717 with a standard error of 0.023, which is not statistically different from the original performance in Table 1, (0.742 with a standard error of 0.034). This indicates that the CPM's performance shows some robustness relative to the specific prompt used.

---

enough for the standard PM to achieve robust performance, while the prompted PM utilizes the capabilities of a generic LLM, trained over a huge dataset.

| Feature name | Attribute | Description |
|---|---|---|
| helpfulness | attribute_desc
attr_min
attr_max | provides valuable assistance to the original poster
no assistance
excellent assistance |
| specificity | attribute_desc
attr_min
attr_max | is detailed and precise
overly vague
highly specific |
| intent | attribute_desc
attr_min
attr_max | accurately grasps the original poster's intent
misinterprets the original poster's intent
perfectly understands the original poster's intent |
| factuality | attribute_desc
attr_min
attr_max | is based on accurate and verifiable information
blatantly incorrect
entirely accurate |
| easy-to-understand | attribute_desc
attr_min
attr_max | is clear and straightforward
extremely difficult to understand
exceptionally easy to understand |
| relevance | attribute_desc
attr_min
attr_max | directs addresses the original poster's query
entirely irrelevant
highly relevant |
| readability | attribute_desc
attr_min
attr_max | is written in a style appropriate for the original poster's level of understanding
extremely difficult to read
exceptionally easy to read |
| enough-detail | attribute_desc
attr_min
attr_max | provides a sufficient level of detail to be helpful
insufficient detail
comprehensive level of detail |
| biased | attribute_desc
attr_min
attr_max | presents an objective and impartial perspective
strong bias or one-sidedness
completely unbiased |
| fail-to-consider-individual-preferences | attribute_desc
attr_min
attr_max | fails to consider the original poster's cultural or individual preferences
fails to consider the original poster's preferences
carefully considers the original poster's preferences |
| repetetive | attribute_desc
attr_min
attr_max | avoids unnecessary repetition
excessively repetitive
not repetitive |
| fail-to-consider-context | attribute_desc
attr_min
attr_max | fails to consider the original poster's situation and background
fails to consider the original poster's context
appropriately considers the original poster's context |
| too-long | attribute_desc
attr_min
attr_max | is concise and avoids unnecessary length
excessively long
appropriately concise |

Table 13: Paraphrased features augmented from the original descriptions in Table 7. Those features are used with the template in Table 5.

