# OpenReview forum: "Compositional Preference Models for Aligning LMs"
_ICLR.cc/2024/Conference — ICLR 2024 poster_

### Official Review · Reviewer_dDjG · 2023-10-27

**Soundness:** 3 good
**Presentation:** 3 good
**Contribution:** 3 good
**Rating:** 6
**Confidence:** 4

**Summary:**

This paper introduces ‘compositional preference models’ (CPMs) that re meant to overcome issues of transparency and scalability found in regular preference models. CPMs decompose preferences into interpretable features, and subsequently aggregates them (with LR). Generally, results show improvement in generalization and avoidance of over-optimization, which are not themselves ’transparency and scalability’.

**Strengths:**

- Although a minor strength, the collection of papers in sections 1, 2, and 6 are sufficient, relatively recent, and relevant.
- The compositional extension, though simple, is technically a novel contribution that appears to provide several benefits.
- The setup for best-of-$n$ sampling seems fair. The full set of 25.6K responses, and code, would be appreciated.
- the use of a separate LLM for evaluation is appreciated.

**Weaknesses:**

- Although somewhat minor, the use of logistic regression will naturally cause some confusion, especially to those who want an end-to-end trainable model for this task. Other models should have been attempted.
- Section 3.1 should be more informative as to the nature of features _c_ and how their set is identified, selected, or defined. This should include both the specific list in Sec 4.1 as well as guidance for other tasks, in general.
- Although another minor weakness, at least one other LLM  should have been used for extraction (e.g., one of the Llamas)
- Very minor, but please correct the various issues with your references including capitalization and completeness (e.g., Amodei suffers from both — use brackets around {AI} and provide full paper details)

**Questions:**

- The definition of ‘model robustness’ in Sec 4.2 seems incomplete — surely a factor is the domain or scenario in which the model is to be deployed or evaluated, too?
- Would it be possible to re-arrange the columns of Fig2a and Fig2b so Standard comes left-most (first)?\
- Would there be value in actually performing human evaluations, despite the findings of best-of-n sampling in related work?

---

> ### Author Response · Authors · 2023-11-21
> **Response to Reviewer dDjG**
>
> We sincerely thank the reviewer for the feedback and constructive comments. We hope that our responses to your questions address most concerns you may have and strengthen the contribution of our work.
>
> Strengths:
> > The full set of 25.6K responses, and code, would be appreciated.
>
> Thanks for your suggestion! We updated the supplementary file.
>
> Weaknesses:
> > Although somewhat minor, the use of logistic regression will naturally cause some confusion, especially to those who want an end-to-end trainable model for this task. Other models should have been attempted.
>
> We appreciate your feedback regarding the selection of the machine learning model.
> Our decision to employ logistic regression was in order to favor the interpretability and intuitiveness of CPM.
> Existing literature supports the effectiveness of preference prediction using a linear combination of scores [1, 2].
>
> Although there are alternative models for preference, we believe logistic regression is the most natural and well-studied general method that can be employed for this purpose.
>
> > Section 3.1 should be more informative as to the nature of features c and how their set is identified, selected, or defined. This should include both the specific list in Section 4.1 as well as guidance for other tasks, in general.
>
> We appreciate your feedback on the impact of prompts and features. We have incorporated your suggestion by adding a general principle for defining features in Section 3.1. In general, the features should be diverse enough to cover a broad range of preferences, while avoiding redundancy to maintain efficiency and interpretability.
>
> In addition, we further investigated the impact of feature description to the CPM's performance by employing GPT-3.5 to paraphrase the original descriptions in Table 7 (For details of this experiment, please refer to Appendix F). The average win-rate (described in Section 4.5) of CPM-Flan-T5 using these augmented prompts was $0.717$ with a standard error of $0.023$, statistically similar to the original performance in Table 1 ($0.742$ with a standard error of $0.034$). This implies that the CPM's performance is not too sensitive to the specific wording of the prompts.
>
> > Very minor, but please correct the various issues with your references including capitalization and completeness (e.g., Amodeisuffers from both — use brackets around {AI} and provide full paper details)
>
> We updated the references, thanks for the suggestion!
>
> Questions:
> > The definition of ‘model robustness’ in Sec 4.2 seems incomplete — surely a factor is the domain or scenario in which the model is to be deployed or evaluated, too?
>
> We totally agree that the overall quality of a preference model depends very much on the scenario in which it will be exploited, but in this section we focus on a narrower notion of robustness, which we can evaluate independently of its downstream usage, namely to what extent the PM is sensitive to the selection of data inside a given preference dataset. Such a notion of robustness is a necessary (although not sufficient) condition for a reliable application of the PM to the downstream task, and section 4.2 argues that the CPM is superior to a standard PM on this dimension.
>
> > Would it be possible to re-arrange the columns of Fig2a and Fig2b so Standard comes left-most (first)?
>
> Thank you for the suggestion, we updated the figures.
>
> > Would there be value in actually performing human evaluations, despite the findings of best-of-n sampling in related work?
>
> Thanks for raising this point about human evaluation. We acknowledge this concern, but we note that it is becoming standard practice to use LLM eval as a proxy for human eval and that it has been shown to be close to human raters for various tasks, e.g. [3, 4, 5, 6]. In addition, for complex alignment tasks such as the ones we consider here, human evaluation is a difficult target to aim for and difficult to reproduce, hindering stable evaluation (see [2, 7]).
>
> [1] SHARMA, Mrinank, et al. Towards Understanding Sycophancy in Language Models. arXiv preprint arXiv:2310.13548, 2023.
>
> [2] HOSKING, Tom; BLUNSOM, Phil; BARTOLO, Max. Human Feedback is not Gold Standard. arXiv preprint arXiv:2309.16349, 2023.
>
> [3] Rafailov, Rafael, et al. Direct Preference Optimization: Your language model is secretly a reward model, in Thirty-seventh Conference on Neural Information Processing Systems, 2023.
>
> [4] LIU, Tianqi, et al. Statistical rejection sampling improves preference optimization. arXiv preprint arXiv:2309.06657, 2023.
>
> [5] LEE, Harrison, et al. RLAIF: Scaling reinforcement learning from human feedback with ai feedback. arXiv preprint arXiv:2309.00267, 2023.
>
> [6] ZHENG, Lianmin, et al. Judging llm-as-a-judge with mt-bench and chatbot arena. CoRR, abs/2306.05685, 2023. doi: 10.48550. arXiv preprint arXiv.2306.05685.
>
> [7] CHIANG, Cheng-Han; LEE, Hung-yi. Can Large Language Models Be an Alternative to Human Evaluations?. arXiv preprint arXiv:2305.01937, 2023.

---

### Official Review · Reviewer_pUSN · 2023-10-28

**Soundness:** 3 good
**Presentation:** 3 good
**Contribution:** 3 good
**Rating:** 6
**Confidence:** 4

**Summary:**

This paper proposes Compositional Perference Models (CPMs), a new perferene model framework that decomposes one global perference assessment into several interpretable features, using a prompted LM to score these features, and finally aggregates these features together with their scores using a logistic regression classifier. Experiments show that CPMs improves generalization and robustness than standard PMs. The main contributions include: (1) new CPMs that allows more transparent supervision; and (2) better results at dimensions of model/overoptimization robustness, generalization, and perference alignment.

**Strengths:**

(1) new CPMs that allows more transparent supervision;
(2) better results at dimensions of model/overoptimization robustness, generalization, and perference alignment.

**Weaknesses:**

1. prefer to see detailed investigations of applying CPMs to different stages of (1) inference only (2) sft, and (3) peft.
2. not quite clear of the scalability of the usage of current 13 features to novel langauges/tasks, further investigations are preferred.

**Questions:**

1. in Table 5 and Table 6, scores from 1 to 10 are used, and did you try other ranges such as 1 to 5, and how did you decide to use a range of 1 to 10? Also, does different features require different scopes/ranges of scores? In addition, when changing from numbers to words (bad, good, better, best...), how shall the results change?
2. any comparison of between supervised fine-tuning (SFT) and PEFT when using the CPMs? Or, any comparison of the usage of resources under different model alignment frameworks? So, (1) inference only stage controlling, (2) sft, (3) peft, any investigations on these directions of using CPMs?
3. page 3, there are 13 features used, any detailed analysis of overlapping or diversities among these features?or when applying your method to other languages/tasks, how shall we reuse these features or how shall we design new features (any common rules?)

---

> ### Author Response · Authors · 2023-11-21
> **Response to Reviewer pUSN, part1**
>
> We are grateful for the reviewer's valuable feedback and suggestions, particularly those regarding the scalability of the current features and potential variations. We believe that our response addresses your main concerns, enhancing the significance of our contribution.
>
> > Prefer to see detailed investigations of applying CPMs to different stages of (1) inference only (2) sft, and (3) peft.
>
> > Not quite clear of the scalability of the usage of current 13 features to novel languages/tasks, further investigations are preferred.
>
> We attempt to address these concerns in our responses to questions 2 and 3 below.
>
> Questions:
>
> > in Table 5 and Table 6, scores from 1 to 10 are used, and did you try other ranges such as 1 to 5, and how did you decide to use a range of 1 to 10? Also, does different features require different scopes/ranges of scores? In addition, when changing from numbers to words (bad, good, better, best...), how shall the results change?
>
> Thanks for highlighting the effect of various prompts. For the score of the feature values, we performed  normalization for each feature to have mean 0 and standard deviation 1, so that the effect of range remains minimal.
>
> To further investigate the impact of various prompts and the robustness of the CPM's performance on prompts, we employed GPT-3.5 to paraphrase the original description in Table 7. The paraphrased features possess similar meaning but different descriptions. The average win-rate (the metric described in Section 4.5) of CPM-Flan-T5 using this paraphrased prompt is $0.717$ with a standard error of $0.023$, which is not statistically different from the original performance in Table 1, ($0.742$ with a standard error of $0.034$). This further indicates that the CPM's performance is robust relative to the specific prompt used. Please see Appendix F for the details of this extended experiment.
>
> > any comparison of between supervised fine-tuning (SFT) and PEFT when using the CPMs? Or, any comparison of the usage of resources under different model alignment frameworks? So, (1) inference only stage controlling, (2) sft, (3) peft, any investigations on these directions of using CPMs?
>
> In order to address this question, for instance at the level of resources used at inference time, we would need to consider a full generative process built on top of the preference model that we propose. This could be done in different ways, for instance by using a PPO-style RLHF fine-tuning of the pretrained model, or in a BoN-style reranking of samples from the pretrained model. Different choices of this kind would have very different consequences on the computational costs at training time (large for the PPO-style fine-tuning, low for the BoN-style sampling) and inference time (low for the PPO-style fine-tuning, large for the BoN-style sampling). But this would certainly be an important question to address in follow-up research.

---

> > ### Author Response · Authors · 2023-11-21
> > **Response to Reviewer pUSN, part2**
> >
> > > page 3, there are 13 features used, any detailed analysis of overlapping or diversities among these features?or when applying your method to other languages/tasks, how shall we reuse these features or how shall we design new features (any common rules?)
> >
> > Thanks for suggesting looking into the diversity of features. To consider this, we computed the correlations between features, see the added Fig. \label{fig:feature-correlation-large} in the Appendix. The figure shows that the features have mostly positive correlations, some of them addressing similar dimensions.
> >
> > In order to figure out the effect of this overlap on the performance of the CPM, we ordered the features based on their importance (i.e. absolute value of the coefficient), and then assessed how the performance of the CPM  — measured in terms of ‘win-rate’ quality as in Section 4.5 of the submission — varies with $k$ when we keep only the first $k$ most important features. Note that regardless of its coefficient rank, we put ‘helpfulness’  first in the ordered list, so that we can compare the case of “Prompted PM with one holistic feature” (namely “helpfulness”) vs “Compositional PM with $k$ features” (see also Table 12 in the Appendix):
> >
> > | Number of features $k$ | Win Rate |
> > | --- | --- |
> > | $k=1$ | 0.707 (0.030) |
> > | $k=3$ | 0.715 (0.024) |
> > | $k=6$ | 0.754 (0.038) |
> > | $k=10$ | 0.735 (0.037) |
> > | $k=14$ | 0.742 (0.034) |
> >
> > The Table 12 shows that the performance of CPM with $k=14$ is worse than that of CPM with $k=6$, related to feature overlap. However the diversity of the features has a positive effect with the performance at $k=1$ falling short of using the combination of all features. Note that the performance gap between $k=14$ and $k=6$ is small, as we employ a regularization term when fitting the logistic classifier.
> >
> > Concerning the design of new features, we admit that this would require additional human intervention, aiming for full coverage and diversity while avoiding overlaps. One approach could be to ask a strong LLM to propose descriptions of the characteristics making one utterance preferable to another one, a form of "LLM-based exploration of underlying features", a further-work direction that we had suggested in our Conclusion.
> >
> > On the other hand, concerning the question of “reuse” of existing features, it should be noted that the CPM approach has one key advantage: these same features can be exploited in different situations, namely over different preference datasets, expressing different possible variants of the notion of helpfulness or of some other alignment criterion. The logistic regression training can be directly applied to such datasets, resulting in a different set of coefficients and ordering of important features.

---

### Official Review · Reviewer_zwab · 2023-10-30

**Soundness:** 2 fair
**Presentation:** 3 good
**Contribution:** 2 fair
**Rating:** 3
**Confidence:** 3

**Summary:**

The paper introduces compositional preference models (CPMs), a new method for aligning language models (LMs) with human preferences. CPMs break down preference scores into multiple features to improve robustness and generalization. This decomposition is accomplished by prompting an LM to assign a value to the answer based on a specific preference type. Experimental findings demonstrate that CPMs effectively mitigate overoptimization in preference modeling.

**Strengths:**

1. Modeling human preferences from different types of judgments is a promising research topic.
2. Experimental results demonstrate that the suggested CPMs indeed improve both robustness and generation.
3. The paper is generally easy to read.

**Weaknesses:**

1. Although CPMs offer a practical method for breaking down preferences by stimulating LMs, I consider it too simplistic and unrealistic to capture intricate human preferences. For instance, easy-to-understand answers and answers with enough details may contradict each other. I have reservations about whether logistic regressors can accurately represent this intricate pattern.
2. In terms of the experimental setup, CPMs prompt much more than standard PM, which raises concerns about their comparability. I recommend that the author include another standard PM baseline that uses a similar prompt budget as CPMs. For instance, prompting the LM $n$ times (where $n$ represents the number of pre-defined preference types for CPMs) through sampling and selecting the final preference score via majority voting.

**Questions:**

1. How is the encoder for $x$ parameterized in logistic regression?

---

> ### Author Response · Authors · 2023-11-21
> **Response to Reviewer zwab, part1**
>
> First, thank you for your appreciation of a number of aspects of our submission! We try to address your main concerns below, and also the Question that you ask.
>
> > Although CPMs offer a practical method for breaking down preferences by stimulating LMs, I consider it too simplistic and unrealistic to capture intricate human preferences. For instance, easy-to-understand answers and answers with enough details may contradict each other. I have reservations about whether logistic regressors can accurately represent this intricate pattern.
>
> You are right that certain preference aspects may in principle be in opposition, and in such cases, the human annotators, when having to choose among two responses which one they prefer, have to make a delicate decision.The logistic regression training technique (i.e., determining the lambda parameters) only attempts to fit the annotator preferences as they are expressed in the training data, and is known for its robustness to combining features that may be correlated, positively or negatively: such dependencies typically have little impact on the predictive power of the method (ability to predict the human preferences, our main focus), even if, in the presence of very high correlations (or anti-correlations), different lambda vectors may produce the same predictions (“parameter non-identifiability”) (see [1]).
>
> On this last point, and also to assess whether we could actually detect strongly conflicting features in our experiments, we computed correlations between features, see the added Fig. \label{fig:feature-correlation-large} in the Appendix. This figure shows, for the features and dataset under consideration, mostly positive correlations between features (including “easy-to-understand” and “enough-detail”), and no higher correlation than 0.84.
>
> In summary, the intricacy of combining different dimensions  is delegated to the human annotators producing the preference training set, with the logistic regression classifier trying to reproduce the preferences displayed in such annotations, and being quite effective at combining the features provided as inputs (see [2] for a related point). Of course, the quality of the predictions depends on that of the extracted features — whether they do correspond to the dimensions actually considered by the annotators (see [3] for a cautionary study) — but this is orthogonal to the effectiveness of logistic regression as a combination mechanism.
>
> [1] HARRELL, F.E., Regression Modeling Strategies, Springer Series in Statistics, 2015.
>
> [2] SHARMA, Mrinank, et al. Towards Understanding Sycophancy in Language Models. arXiv preprint arXiv:2310.13548, 2023.
>
> [3] HOSKING, Tom; BLUNSOM, Phil; BARTOLO, Max. Human Feedback is not Gold Standard. arXiv preprint arXiv:2309.16349, 2023.

---

> > ### Author Response · Authors · 2023-11-21
> > **Response to Reviewer zwab, part2**
> >
> > > In terms of the experimental setup, CPMs prompt much more than standard PM, which raises concerns about their comparability. I recommend that the author include another standard PM baseline that uses a similar prompt budget as CPMs. For instance, prompting the LM n times (where n represents the number of pre-defined preference types for CPMs) through sampling and selecting the final preference score via majority voting.
> >
> > In order to try and address this concern to some extent, we have ordered the features based on their importance (i.e. absolute value of the coefficient), and then assessed how the performance of the CPM (here CPM-Flan-T5)  — measured in terms of ‘win-rate’ quality as in section 4.5 of the submission — varies with $k$ when we kept only the first $k$ most important features. One exception to that ordering is that (regardless of its coefficient rank) the holistic feature ‘helpfulness’ comes first in the ordered list, so that we can compare the case of “Prompted PM with one holistic feature” (namely “helpfulness”) vs “Compositional PM with $k$ features”, in a simple, yet illustrative, way.
> > The ordered feature list is: `['helpfulness', 'fail-to-consider-context', 'enough-detail', 'factuality', 'length',' biased', 'easy-to-understand', 'specificity', 'too-long', 'intent' , 'repetitive', 'fail-to-consider-individual-preferences', 'relevance', 'readability']`, and we obtain the following win-rates (averaged for 5 trials + standard error), for a representative subset of $k$’s (see also Table 12 in the Appendix):
> >
> > | Number of features $k$ | Win Rate |
> > | --- | --- |
> > | $k=1$ | 0.707 (0.030) |
> > | $k=3$ | 0.715 (0.024) |
> > | $k=6$ | 0.754 (0.038) |
> > | $k=10$ | 0.735 (0.037) |
> > | $k=14$ | 0.742 (0.034) |
> >
> > This experiment suggests two initial observations. First, the single holistic feature ‘helpfulness’ obtains a reasonable win-rate (0.707) on its own, but falls short of using the combination of all features. Second, while it is true that computing 14 features incurs a larger cost than just computing one, it is actually possible to obtain similar (or even slightly better) results by keeping only the first 6 features. Much more sophisticated techniques [1] for logistic regression ‘feature selection’ than this one could certainly be employed to limit the number of features required without hurting the performance of CPMs.
> >
> > While this experiment illustrates the possibility of lowering the computing budget by concentrating on the most significant features, there is however another aspect to your concern, namely the possibility of designing a powerful unique holistic prompt for helpfulness that would be as effective as the several prompts that we use. While prompting is not the standard way PMs are implemented, this would indeed be a possibility, where one would focus on an experimental search for such a powerful prompt, and that could be an interesting topic on its own — as would also be an experimental search for more powerful prompts for each of the underlying dimensions that we consider in the CPM, which our current experiments suggest would collectively beat a holistic prompt.
> >
> > > How is the encoder for x parameterized in logistic regression?
> >
> > Thank you for this question! We have updated section 3.1 to clarify that each feature score depends on $x$, which is part of the prompt, as well as section 3.2 to clarify that the logistic regression takes as input these feature scores.
> >
> > [1] HARRELL, F.E., Regression Modeling Strategies, Springer Series in Statistics, 2015.

---

> ### Comment · Reviewer_zwab · 2023-11-22
>
> Thank you for your response.
>
> > First, the single holistic feature ‘helpfulness’ obtains a reasonable win-rate (0.707) on its own.
>
> I am not sure how to interpret this. What is the reason behind the considerably higher win-rate compared to `Standard PM` (0.707 vs 0.588)?

---

> > ### Author Response · Authors · 2023-11-22
> >
> > Thank you for your acute observation!
> > We would like to emphasize the distinction between the prompted PM with holistic feature ‘helpfulness’ and the standard PM. The prompted PM utilizes the capabilities of a generic LLM, trained over a huge dataset, while the standard PM is trained on a specific dataset, typically much smaller.
> > We hypothesize that the superior performance of the holistic PM over the standard PM here is due to the fact that the preference dataset size of 20K may be insufficient for the standard PM to achieve robust performance. While the CPM — now based on several prompted features — is also trained on the same dataset, its training is limited to the combination of features through logistic regression, requiring less data than the standard PM, as illustrated by Figures 2 and 5 in the main text.
> > We will update the discussion of Table 12 in the submission to reflect your observation and our above comments.

---

### Official Review · Reviewer_Q5ds · 2023-10-31

**Soundness:** 3 good
**Presentation:** 4 excellent
**Contribution:** 3 good
**Rating:** 8
**Confidence:** 4

**Summary:**

The paper presents a compositional preference model for LLM alignment. In contrast to standard monolithic preference models that assign a single scalar value to preference judgments. the model uses a number of features associated with individual scalar values (assigned by an LLM as an automatic evaluator) that are then linearly combined into an overall score.  The authors argue that this provides an inductive bias to the model that makes it more robust to overfitting and reward hacking and results in better generalization and human interpretability. The technique is evaluated with respect to consistency of responses for models trained on different subsets of the training data), comparison against reference PMs from the literature, robustness to overoptimization, and alignment of LLMs trained with the proposed model as opposed to a standard PM.

**Strengths:**

The core idea of the paper is simple yet powerful and addresses known weaknesses in traditional monolithic preference models; it should be of broad interest to the ICLR audience. The presentation is clear and -- with the exception of human alignment evaluation (see below) -- the evaluations are convincing.

**Weaknesses:**

For alignment with human preferences, another  LLM (Claude-2) was used rather than genuine human ratings. Although there is more effort associated with a human evaluation study,  and the literature you cite has shown some (imperfect) degree of correlation between human ratings and LLM scores, I really consider human evaluation a must here - otherwise, you are measuring alignment between different LLMs, which can simply result from similar training procedures or similar preference models used.

**Questions:**

1. It looks like the feature evaluator LLMs (Flat-T5 and GPT3.5) were used out of the box with prompting for assigning feature scores, without fine-tuning or in-context learning. I would have like to see the comparison against fine-tuned versions for each feature.
2. How does the robustness of the CPM change with an increasingly larger list of features?

---

> ### Author Response · Authors · 2023-11-21
> **Response to Reviewer Q5ds, part1**
>
> Thank you for your detailed and thoughtful review. We are glad you found our paper well-written and interesting. Please find responses to your questions below. We hope that our answers address your main concerns, increasing the strength of our contribution.
>
> > For alignment with human preferences, another LLM (Claude-2) was used rather than genuine human ratings. Although there is more effort associated with a human evaluation study, and the literature you cite has shown some (imperfect) degree of correlation between human ratings and LLM scores, I really consider human evaluation a must here - otherwise, you are measuring alignment between different LLMs, which can simply result from similar training procedures or similar preference models used.
>
> Thanks for raising this point about human evaluation. We acknowledge this concern, but we note that it is becoming standard practice to use LLM eval as a proxy for human eval and that it has been shown to be close to human raters for various tasks, e.g. [1, 2, 3]. This said, we believe that, for complex modeling tasks as the ones we consider here, human evaluation is a difficult target to aim for, as it often depends on informal criteria that are not defined precisely (see [4] on a related point). One radical way to escape this dilemma would be to assume that human preferences are fully represented by collecting a dataset of preference judgments obtained based on a certain protocol, and that the quality of a model would be assessed through its ability to predict new preference judgments based on the same protocol, which is similar with what we do with our robustness studies. However, this argument is not without some circularity, but it is not obvious to us how to escape it.
>
> [1] Rafailov, Rafael, et al. Direct Preference Optimization: Your language model is secretly a reward model, in Thirty-seventh Conference on Neural Information Processing Systems, 2023.
>
> [2] LIU, Tianqi, et al. Statistical rejection sampling improves preference optimization. arXiv preprint arXiv:2309.06657, 2023.
>
> [3] LEE, Harrison, et al. RLAIF: Scaling reinforcement learning from human feedback with ai feedback. arXiv preprint arXiv:2309.00267, 2023.
>
> [4] HOSKING, Tom; BLUNSOM, Phil; BARTOLO, Max. Human Feedback is not Gold Standard. arXiv preprint arXiv:2309.16349, 2023.

---

> > ### Author Response · Authors · 2023-11-21
> > **Response to Reviewer Q5ds, part2**
> >
> > Questions:
> > > It looks like the feature evaluator LLMs (Flat-T5 and GPT3.5) were used out of the box with prompting for assigning feature scores, without fine-tuning or in-context learning. I would have like to see the comparison against fine-tuned versions for each feature.
> >
> > Thank you for proposing these interesting  alternatives to our feature evaluator: fine-tuning or in-context learning. However, we believe such approaches would be challenging in our case, due to the difficulty of acquiring ground truth feature scores for each feature, a problem which would exist in both alternatives (but less severely in the ICL case).
> >
> > > How does the robustness of the CPM change with an increasingly larger list of features?
> >
> > In order to address this question to some extent, based on our existing list of features, we ordered these features based on their importance (i.e. absolute value of the coefficient), and then assessed how the performance of the CPM (here CPM-Flan-T5)  — measured in terms of ‘win-rate’ quality as in section 4.5 of the submission — varies with $k$ when we kept only the first $k$ most important features to that ordering [one exception, related to our response to a comment of reviewer zwab, is that (regardless of its coefficient rank) the holistic feature ‘helpfulness’ comes first in the ordered list].
> >
> > The ordered feature list is: `['helpfulness', 'fail-to-consider-context', 'enough-detail', 'factuality', 'length',' biased', 'easy-to-understand', 'specificity', 'too-long', 'intent' , 'repetitive', 'fail-to-consider-individual-preferences', 'relevance', 'readability']`, and we obtain the following win-rates (averaged for 5 trials with standard error), for a representative subset of $k$’s (see also Table 12 in the Appendix):
> >
> > | Number of features $k$ | Win Rate |
> > | --- | --- |
> > | $k=1$ | 0.707 (0.030) |
> > | $k=3$ | 0.715 (0.024) |
> > | $k=6$ | 0.754 (0.038) |
> > | $k=10$ | 0.735 (0.037) |
> > | $k=14$ | 0.742 (0.034) |
> >
> > This experiment suggests that the quality of the CPM increases significantly with the number of features considered. However, interestingly, for that experiment, stopping at the 6 first features (ie, the 5 most important ones plus the “helpfulness” feature) obtains higher quality than using all the available features.
> > Of course this experiment only scratches the surface of your question. In order to answer more completely, one would have to use more sophisticated techniques for feature selection in logistic regression models. It is to be noted that, according to specialized literature on linear and logistic regression, e.g. [1], adding more features can in principle only improve the predictive quality of the model, assuming enough training data is available. In other words, from the point of view of predictivity, the main problem with many features is the risk of overfitting. On the other hand, orthogonal to predictivity, highly correlated features may lead to difficulty in uniquely identifying the coefficients of the model, but that is a different question (see also our response to zwab).
> >
> > [1] HARRELL, F.E., Regression Modeling Strategies, Springer Series in Statistics, 2015.

---

> ### Comment · Reviewer_Q5ds · 2023-11-21
>
> I'm aware of the problems with human preferences as ground truth (acquisition, reliability, etc.), but as you say, your argument is circular. I suggest defining a controlled scenario where preferences can be collected more easily, or coupling preference modeling with an end-to-end task (e.g., helpfulness in the context of a task-oriented dialog scenario where success (task accomplishment) can be measured more easily. Agree that this may exceed the scope of the paper, but in that case I'd avoid the term 'alignment with human preferences' and include a few sentences of discussion of the limitations.
> Thank you for the added data points on number and combination of features, this addresses my question.

---

> > ### Author Response · Authors · 2023-11-22
> >
> > We appreciate your feedback and additional suggestions regarding the controlled scenario. The primary focus of our paper is on complex modeling tasks where decomposing a preference (e.g., "is this text preferable?") into a series of simpler features (e.g., "is this text easy to understand?", "is this text informative?") is beneficial for evaluating language models and enabling human oversight. While a controlled scenario such as the one you propose is possible, it falls outside the scope of our current work.
> > Nevertheless, we believe that exploring coupled preference modeling would be a valuable direction for future research. Additionally, we agree with the potential for human evaluation to guide preference modeling. We will address these points in the conclusion, acknowledging the limitations of our approach and proposing directions for future work.

---

### Author Response · Authors · 2023-11-23
**General Thanks and Response to All Reviewers**

We thank all the reviewers for their very constructive feedback and useful suggestions for improvements. Our detailed answers are provided in responses to individual reviews, but some main themes emerge.
We clarify the validity of the logistic regression approach based on relevant literature, conduct additional experiments showing that composing individual features yields better results than relying on a single holistic feature, but that too many features can lead to overlap, and try to provide some guidance for feature construction and check that the performance of CPM is robust to the formulation of the prompt used for characterizing each feature.
These improvements are highlighted in cyan-colored text throughout the revised paper.
We believe that these changes will make it more valuable to the community.

---

### Meta-Review · Area_Chair_V1se · 2023-12-07

**Metareview:**

The submission introduces an approach called compositional preference models (CPMs) which expresses the learned reward as a weighted sum of scalar features obtained from prompting a (pretrained, frozen) language model to provide a score for the input sequence on various aspects of quality. The paper considers 13 features such as helpfulness, specificity, readability, etc. and trains CPM's weighting of each feature on a dataset of pairwise preferences as usual, i.e., by using the reward model as part of a Bradley-Terry model. The inductive bias in CPM is argued to make the reward model more robust to overfitting and reward hacking, in addition to making it more interpretable.

Reviewers note the proposed approach is a straightforward yet powerful idea (Q5ds, dDjG) supported by convincing experimental results (zwab, pUSN) and is of broad interest to the ICLR audience (Q5ds). Reviewer zwab also notes the paper's clarity, and Reviewer dDjG notes its comprehensive related work discussion. The main reviewer concerns are:

- Reviewer Q5ds is unsatisfied with the paper's use of another LLM rather than genuine human ratings for evaluation. The authors acknowledge this is a concern but note that this is becoming standard practice. Both Reviewer Q5ds and the authors agree this is not an ideal situation, but that addressing this shortcoming is beyond the scope of the paper, and Reviewer Q5ds is still positive on acceptance despite that.
- Reviewer dDjG's concern over the exclusive use of logistic regression is addressed to their satisfaction by the authors' response, and so is their concern over the organization of Section 3.1. As a result, their score would lie at a 7 if such a score was allowed.
- Reviewer zwab's concerns over the simplicity of logistic regression standing in the way of appropriately capturing human preferences and over the approach's added LM prompting requirements are adequately addressed from my perspective. The authors rightfully point out that the logistic regression classifier is empirically quite effective, to which I would add that the features the classifier is trained on (obtained by prompting a sophisticated LM) are themselves fairly sophisticated; the Reviewer's objection would be akin to objecting that linear probing for transfer learning cannot possibly capture the conditional distribution of the labels given the inputs because a softmax classifier is a simple classifier. The authors also break down the win rate of CPM over regular preference modeling as a function of the number k of features used, showing that even if only the "helpfulness" feature is used the resulting CPM still comes out ahead in terms of win-rate. This is convincing to me.
- Reviewer pUSN's questions appear to have been appropriately answered by the authors, as is their concern over the scalability of CPM in terms of number of features used by the logistic regression classifier.

Most reviewers feel positive about acceptance, and the concerns expressed by the one reviewer against acceptance have been addressed by the authors from my vantage point. I recommend accepting the paper.

**Justification For Why Not Higher Score:**

There is good support for acceptance, but the evaluation's use of other LLMs rather than real human preferences gets in the way of a stronger recommendation.

**Justification For Why Not Lower Score:**

Most reviewers feel positive about acceptance, and the concerns expressed by the one reviewer against acceptance have been addressed by the authors from my vantage point.

---

### Decision · Program_Chairs · 2024-01-16

Accept (poster)